

# Predicting Geomagnetic Indices for Space Weather Applications in Solar Cycle 25

Mu He[1], Hongbing Zhu[2]

[1]College of Artificial Intelligence, Suzhou Chien-Shiung Institute of Technology, Suzhou, 215411, Jiangsu, China
[2]Avant-courier Laboratory, 736-0067, Hiroshima, Japan

*Correspondence to*: Hongbing Zhu (chinmumu@gmail.com)

**Abstract.** This study investigated the relationship between geomagnetic indices (Ap and DST) and solar activity with non-metric multidimensional scaling (NMDS) and the novel LSTM+ forecasting model. The NMDS analysis revealed a stronger association of Ap with overall solar activity and solar wind conditions compared to DST, highlighting the influence of elevated
plasma flow speed and proton temperature on geomagnetic disturbances. The LSTM+ model, incorporating a dynamic reforecast procedure, demonstrated high accuracy in predicting Ap and DST, achieving strong performance metrics for SC-24. Based on the model and historical trends, the peak Ap and trough DST for SC-25 are projected to occur between May 2026 and January 2027, aligning with the observed lag between sunspot number and geomagnetic indices. These findings enhance our understanding of solar-terrestrial interactions and provide a valuable tool for space weather prediction, crucial for
mitigating potential impacts on technological infrastructure.

## 1 Introduction

The magnetosphere of the Earth is perpetually influenced by the dynamic effects of solar phenomena, with the solar wind and coronal mass ejections (CMEs) serving as principal catalysts for geomagnetic disturbances (Kivelson and Russell, 1995). These disturbances, particularly amplified during periods of heightened solar activity known as solar storms, can induce
significant fluctuations in the Earth's magnetic field. A comprehensive understanding of these fluctuations is essential for accurate forecasting and mitigation of space weather impacts. These impacts can have far-reaching consequences for a wide range of systems, including technological infrastructure, human activities in space, and even terrestrial operations (Schrijver et al., 2015).

Quantifying and understanding these geomagnetic variations are paramount for space weather prediction and mitigation efforts.
Two key indices employed for this purpose are the Ap index and the Disturbance Storm Time (DST) index. The Ap index, a widely used indicator of global geomagnetic activity at mid-latitudes, reflects both minor and major geomagnetic storms. It is derived from the planetary Kp index, which quantifies geomagnetic disturbances on a quasi-logarithmic scale at three-hour intervals (Mayaud, 1980). Conversely, the DST index functions as a gauge for the intensity of the ring current that circulates around the Earth during geomagnetic storms, resulting in a diminishment of the horizontal component of the geomagnetic field,





particularly in the equatorial region (Mayaud, 1980). Negative Dst values signify the occurrence of a geomagnetic storm, with lower values denoting more severe storm conditions (Nose et al., 2015). The Ap and Dst indices constitute essential parameters for the monitoring and investigation of geomagnetic variations and their correlation with solar activity.

Recent studies have investigated the use of deep learning and data assimilation techniques to enhance Dst forecasting accuracy. For instance, Zhang et al. demonstrated that Long Short-Term Memory (LSTM) networks achieve superior short-term

predictive performance, particularly during geomagnetically quiet periods. Furthermore, they found that incorporating Empirical Mode Decomposition (EMD) with LSTM effectively addresses prediction lag, making the EMD-LSTM model more suitable for forecasting geomagnetic stormss (Zhang et al., 2023). Nilam and Ram demonstrated the high accuracy of the Ensemble Kalman Filter (EnKF) for Dst forecasting, particularly during severe geomagnetic storms, through the assimilation of real-time data (Nilam and Ram, 2022). Significantly, Abduallah et al. introduced the Dst Transformer, a novel Bayesian

deep learning model that surpasses existing methods in both accuracy and uncertainty quantification for Dst forecasting. This study represents the first application of a Bayesian deep learning approach specifically designed for predicting the Dst index (Abduallah et al., 2022). Analysis of Ap index forecasting performance indicates that current Space Weather Prediction Center (SWPC) predictions achieve moderate to strong accuracy for Day-0 forecasts, as evidenced by Pearson's correlation coefficients ranging from 0.57 to 0.79. However, predictive accuracy diminishes considerably for Day-2 forecasts, with

correlation coefficients dropping to a range of 0.37 to 0.44 (Paouris et al., 2021). Linear prediction filters, particularly autoregressive moving average (ARMA) models incorporating past Ap index behavior and solar wind velocity as inputs, have shown promise for enhancing Ap index forecasting accuracy. These models have demonstrated the capability to predict up to 57% of the variance in the Ap index, surpassing the performance of human forecasters in some cases (McPherron, 1999). These findings highlight the need for exploring advanced techniques, like incorporating solar wind parameters, to enhance Ap

index forecasting accuracy, particularly for longer lead times.

In this work, the relationship for DST and Ap indices with solar activity indicators over Solar Cycles 21 to 24 is analyzed. Statistical analysis between geomagnetic indices and solar drivers are examined to provide a comprehensive understanding of their historical interactions. Additionally, the optimized LSTM+ model is trained on the historical values of DST and Ap to forecast their future variations during Solar Cycle 25. The predicted results are compared with the identified relationships from

previous cycles to determine the periods during which the indices are expected to reach their minimum or maximum values. This investigation is anticipated to facilitate the enhancement of predictive frameworks for geomagnetic activity and to augment our insights into the forecasting of space weather phenomena.

## 2 Relationships between geomagnetic and solar parameters

### 2.1 Data source

Data pertaining to the Disturbance Storm Time (DST) index, Ap index, Sunspot Numbers (SSN), F10.7 cm solar radio flux (F10.7), and solar wind parameters (Field Magnitude Average |B|, proton temperature, plasma flow speed, Na/Np ratio, and





flow pressure) were retrieved from the OMNIWeb Plus dataset, curated by NASA's Space Physics Data Facility (SPDF) and available at https://spdf.gsfc.nasa.gov/pub/data/omni/low_res_omni/. Sunspot area (SSA) data were sourced from the Royal Observatory, Greenwich (RGO) USAF/NOAA dataset, accessible online at http://solarcyclescience.com/activeregions.html.

The Solar Flare Index (SFI) was obtained from the composite SFI database version 25, a collaborative effort between the Astronomical Institute Ondřejov Observatory of the Czech Academy of Sciences and the Kandilli Observatory of Istanbul, Turkey (Balch, 2009). To ensure consistency and a comprehensive temporal coverage, all datasets spanning the period from 1976 to 2024 were utilized in this study.

## 2.2 Relationships between geomagnetic and solar parameters

Non-metric multidimensional scaling (NMDS) was employed to investigate the relationship between geomagnetic indices and a suite of solar parameters. Annual averages for each parameter were calculated and standardized using z-scores to ensure data integrity and comparability. Euclidean distance served as the dissimilarity measure, and Young's S-stress formula (Young and Hamer, 2013) guided the optimization process. The iterative process converged after six iterations, achieving a stress value of 0.02727 and an R-squared (RSQ) value of 0.98978, indicative of a strong model fit. The resulting two-dimensional NMDS

solution, depicted in Figure 1, visually represents the spatial relationships between the selected geomagnetic indices and solar parameters. The horizontal axis (Dimension 1) appears to represent the overall level of solar activity, with higher values corresponding to increased activity. The vertical axis (Dimension 2) seemingly captures the nature of solar wind disturbances, with positive values associated with enhanced plasma flow speed and proton temperature, and negative values linked to reduced solar wind pressure and geomagnetic storm intensity.

The NMDS configuration revealed a complex interplay between geomagnetic indices and solar parameters. The DST index, reflecting the intensity of geomagnetic storms, was positioned distally from the cluster of solar activity indicators. This suggests that DST is influenced by a combination of factors rather than individual solar parameters alone. Conversely, the Ap index, representing the overall level of geomagnetic disturbance, was situated closer to the cluster, indicating a stronger association with general solar activity. Specifically, the NMDS solution suggests that elevated plasma flow speed and proton

temperature are linked to higher Ap values, whereas decreased solar wind pressure is associated with lower Ap values. This finding corroborates the hypothesis that high-speed solar wind streams and enhanced proton temperature can drive geomagnetic disturbances. Notably, the closer proximity of solar wind parameters to Ap compared to DST implies a more direct influence on the former. The solar activity indicators, including SSN, SSA, and SFI, formed a distinct cluster in the NMDS configuration, highlighting their close interrelationship. These indicators exhibited a moderate association with Ap,

suggesting that heightened solar activity can contribute to geomagnetic disturbances. However, their relationship with DST was less pronounced, indicating that geomagnetic storms are not solely driven by these specific indicators.





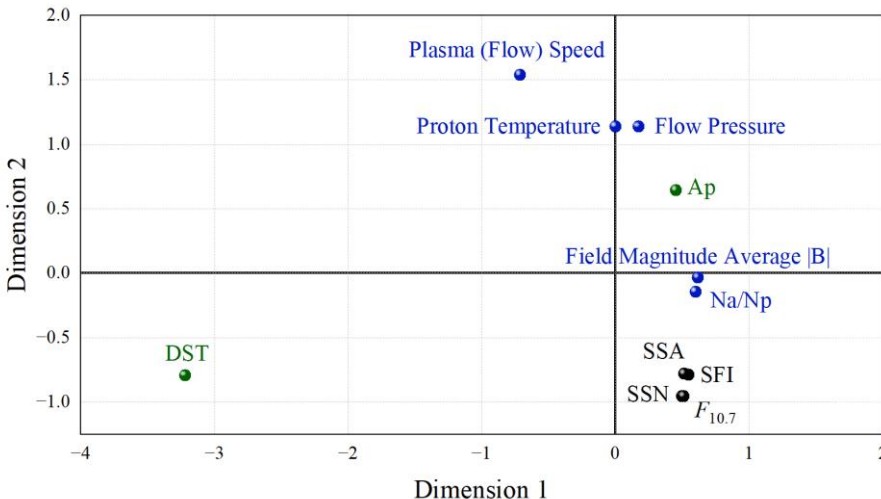

**Figure 1: The result of Non-Metric Multidimensional Scaling.**

## 3 Prediction procedure

### 3.1 The LSTM+ model

This study employs the LSTM+ model, a forecasting framework designed to enhance the predictive accuracy of traditional LSTM networks for solar activity (Zhu et al., 2023, 2022). The LSTM+ model distinguishes itself through the incorporation of a novel reforecast (RF) procedure, which iteratively feeds the most recent forecast output back into the model as input for subsequent predictions. This dynamic approach, unlike the reliance on solely historical data in traditional LSTM models,

enables the LSTM+ model to capture genuine time series trends and adapt to evolving patterns in solar activity. The RF procedure operates by initially generating a forecast from a set of historical data inputs. This initial forecast then becomes an input alongside a shifted window of historical data to produce the subsequent forecast, establishing a chain of predictions that reflects the dynamic nature of the time series. The performance of the LSTM+ model is significantly influenced by the number of units in the LSTM's hidden layer (N) and the batch size (B), both of which require careful optimization to balance model

complexity and computational efficiency with predictive accuracy and training speed. A large N can potentially lead to overfitting and increased training durations, while a small N may restrict the model's ability to capture complex dependencies within the solar activity data. Similarly, small values of B can hinder convergence due to noise in gradient updates, while large values, though computationally efficient, may necessitate more training epochs to achieve optimal accuracy (Wang et al., 2021).





## 3.2 Evaluation Indices

This study focuses on forecasting the AP and DST trends during Solar Cycle 25 (SC-25), with a particular emphasis on accurately predicting the timing of peak activity. To rigorously evaluate the performance of the implemented LSTM+ model, a comprehensive suite of four evaluation metrics was employed. The Nash-Sutcliffe model efficiency coefficient (NS), ranging from negative infinity to 1, was utilized to quantify the overall predictive skill of the LSTM+ model, with a value of 1 signifying a perfect forecast. Furthermore, the accuracy of the predicted peak value and its corresponding timing were assessed using the absolute percentage error of the peak or trough value ($ER$) and the absolute percentage error of the time of peak or trough occurrence $E_T$ were employed, respectively. In addition, the Pearson correlation coefficient ($R$) was calculated to gauge the level of agreement between the predicted and observed trend variations. The mathematical definitions of these evaluation indicators are provided below. $V_{AP}$ denotes the peak value of the actual value, $V_{PP}$ denotes the peak value of the predicted value. And $V_{Ai}$ represents the $i-th$ actual value, and $V_{Pi}$ represents the $i-th$ predicted value. $\overline{V_A}$ is the average of the actual values, and $\overline{V_P}$ is the average of the predicted values. $T_{AP}$ and $T_{PP}$ represent the actual and predicted time value of the peak value in days, respectively.

$$NS = 1 - \frac{\sum_{i=1}^{N}(V_{Ai}-V_{Pi})^2}{\sum_{i=1}^{N}(V_{Ai}-\overline{V_A})^2} \tag{1}$$

$$ER = \frac{|(V_{AP}-V_{PP})|}{V_{AP}} \times 100\% \tag{2}$$

$$E_T = T_{AP} - T_{PP} \tag{3}$$

$$r = \frac{\sum_{i=1}^{n}(V_{Ai}-\overline{V_A})(V_{Pi}-\overline{V_P})}{\sqrt{\sum_{i=1}^{n}(V_{Ai}-\overline{V_A})^2 \sum_{i=1}^{n}(V_{Pi}-\overline{V_P})^2}} \tag{4}$$

## 4 Results and Discussion

To maintain the integrity of historical data and ensure robust predictive performance, the LSTM+ model employed in this study was trained and validated using AP and DST data spanning SC-21 to SC-23. This training dataset was then used to generate predictions for SC-24. For the subsequent prediction of SC-25, the model utilized the parameter combination that demonstrated the most favorable performance across all evaluation indices for the AP and DST predictions, respectively, based on the SC-24 forecasting results.

Prior to model training, a critical preprocessing step involving the normalization of the AP and DST data was implemented. This normalization procedure, conducted independently of the LSTM+ model development, scaled the AP and DST data to a mean of 0 and a standard deviation of 1 within each individual solar cycle. This approach aimed to mitigate the inherent variability in cycle intensities, a factor that can significantly confound long-term forecasting endeavors. By reducing the influence of absolute AP and DST magnitudes, the model was able to focus on discerning the underlying temporal patterns of





geomagnetic activity, such as the timing and shape of peaks, without being unduly biased by variations in peak activity levels
across different cycles. The decision to perform this independent normalization stemmed from the research objective, which prioritizes characterizing the temporal evolution of AP and DST within SC-25, rather than making direct comparisons of absolute intensities across cycles. Therefore, normalization was conducted outside the LSTM+ model framework to ensure that the model's predictions remained focused on capturing the time-dependent trends and cyclical structure inherent in the AP and DST data, facilitating a more accurate and nuanced forecast for SC-25.

Figure 2 presents a comparative analysis of normalized AP and DST indices, spanning SC-21 to SC-24. The figure illustrates both the observed values (represented by a black line) and the model-predicted values for AP (brownish-yellow line) and DST (blue line). A comprehensive evaluation of the prediction accuracy is provided in Table 1, utilizing four distinct performance metrics. Focusing on SC-24, the model demonstrated notable accuracy in predicting the peak value of AP and the trough value DST, with absolute percentage errors of 2.90% and 8.11%, respectively. Although the predicted maximum of the Ap index
occurred 108 days prior to the actual peak and the predicted minimum of the Dst index occurred 81 days earlier than the observed trough, these discrepancies represent relatively minor errors in the context of the 11-year solar cycle duration. Statistical analysis further revealed a strong positive correlation between the predicted and actual values for both indices. The AP prediction achieved an NS of 0.90, indicating a near-perfect model performance. Although the DST prediction exhibited a slightly lower NS of 0.80, this value still falls within an acceptable range, suggesting a good level of predictive accuracy.
Pearson correlation coefficients attained highly significant levels (p < 0.01), reaching 0.960 for AP and 0.941 for DST, signifying a close alignment between the predicted and observed temporal variations.

**Table 1. The results of four evaluation indices for the prediction of SC-24 in the LSTM+ model.**

|  | *NS* | *ER* | $E_T$ (in days) | *R* |
|---|---|---|---|---|
| AP | 0.90 | 2.90% (peak) | -108 (peak) | 0.960** |
| DST | 0.80 | 8.11% (trough) | 81 (trough) | 0.941** |

**. Correlation is significant at the 0.01 level (2-tailed).

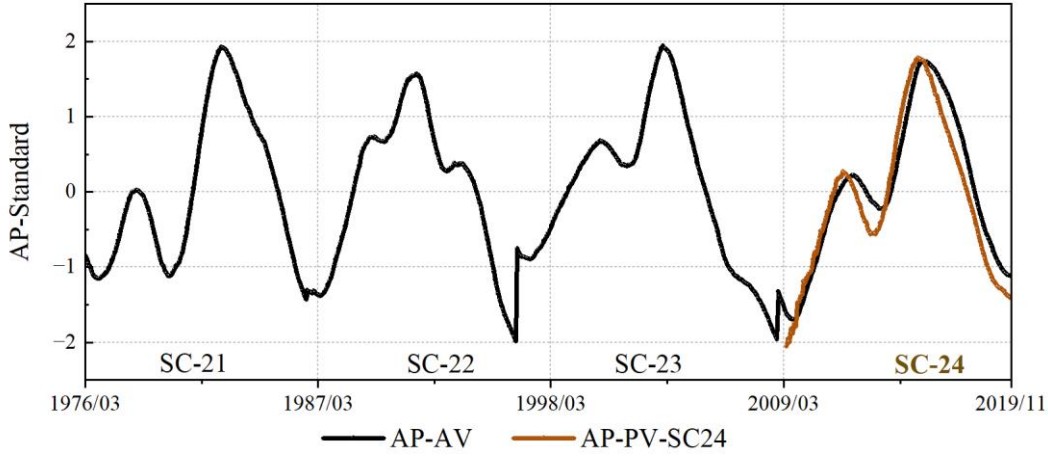



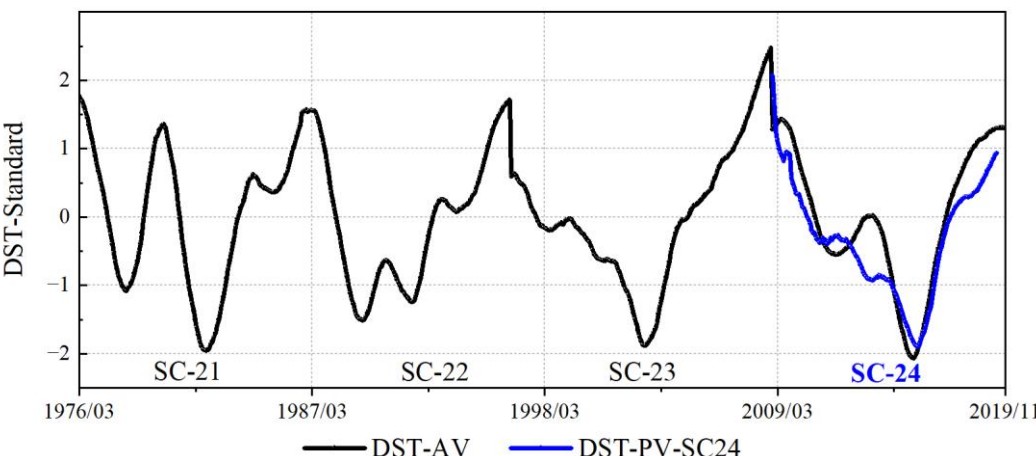


**Figure 2: Normalized AP and DST data from SC-21 to SC-24: actual values (AP-AV, DST-AV) and predicted values for SC 24 (AP-PV-SC24, DST-PV-SC24).**

Figures 3 illustrate the temporal evolution of AP and DST, throughout SC-25. The figures present a comparative analysis of the observed values (denoted as AP-AV and DST-AV) and the predicted values generated by the LSTM+ model (denoted as

AP-PV and DST-PV). Statistical analysis reveals a robust positive correlation between the observed and predicted values for both indices. Pearson correlation coefficients reach 0.894 for AP and 0.917 for DST, both statistically significant ($p < 0.05$). This strong correlation underscores the model's capacity to accurately capture the underlying temporal patterns and trends of these geomagnetic indices. Furthermore, the LSTM+ model successfully identifies the timing of the AP maximum and the DST minimum, both projected to occur in September 2026. This prediction aligns with historical observations across multiple

solar cycles.

Analysis of past annual mean data for AP, DST, and SSN (sunspot number) reveals a consistent tendency that the AP peak typically coincides with the DST trough, both lagging behind the SSN peak by one to three years. Preliminary observations indicate that the unsmoothed monthly average SSN for SC-25 reached its maximum in August 2024. While further data is needed for confirmation, this potentially represents the ultimate peak for the current solar cycle. Notably, our model's

prediction of an AP peak and DST trough in September 2026 falls within two years of this observed SSN peak, well within the established historical lag range.

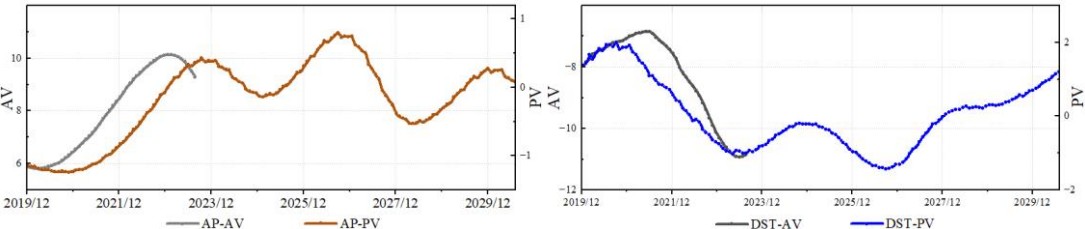

**Figure 3: Comparison of the actual values of AV (AP-AV) and DST (DST-AV) in SC-25 the predicted values (AP-PV, DST-PV) obtained from the LSTM+ Model, respectively.**




## 5 Conclusion

This study investigated the relationships between geomagnetic indices (Ap and DST) and an array of solar parameters using non-metric multidimensional scaling (NMDS) and employed the LSTM+ model to forecast Ap and DST throughout SC-25.

The NMDS analysis revealed a complex relationship between geomagnetic indices and solar parameters. The geomagnetic index AP is more closely associated with overall solar activity and solar wind parameters compared to the geomagnetic index DST. High-speed solar wind streams, enhanced proton temperature, and increased solar activity contribute to geomagnetic disturbances, as reflected in AP. However, the development of geomagnetic storms, as indicated by DST, appeared to be influenced by a confluence of factors rather than being solely attributable to individual solar parameters. Further investigation

is warranted to fully elucidate the interplay of these factors and their implications for the Earth's magnetosphere.

The LSTM+ model, enhanced by a novel reforecast procedure, demonstrated considerable predictive capabilities. Following training and validation with data spanning SC-21 to SC-23, the model successfully predicted the peak Ap value and the trough DST value for SC-24, achieving high Nash-Sutcliffe efficiency coefficients (0.90 and 0.80, respectively) and strong Pearson correlations (0.960 and 0.941, respectively). Considering the 24-month Gaussian smoothing applied to the data and the LSTM+

model's accuracy in predicting SC-24 (within approximately ±4 months), the maximum Ap value and the minimum DST value for SC-25 are anticipated to occur between May 2026 and January 2027. This projection aligns with historical observations of a one-to-three-year lag between the peak in sunspot number (SSN) and the peaks/troughs of Ap and DST. Preliminary observations suggest that the unsmoothed monthly average SSN for SC-25 peaked in August 2024, placing the model's prediction for Ap and DST within the established historical lag range.

These findings contribute significantly to our understanding of the intricate relationship between solar activity and geomagnetic fluctuations. The LSTM+ model, with its ability to capture dynamic temporal trends, emerges as a valuable tool for forecasting geomagnetic indices, crucial for space weather prediction and mitigating potential impacts on technological infrastructure and human activities. Future research avenues could explore incorporating additional solar and geophysical parameters into the model to further enhance its predictive accuracy. Moreover, a deeper investigation into the physical

mechanisms underlying the observed relationships between solar activity and geomagnetic disturbances would provide valuable insights into the Sun-Earth connection.

Competing interests: The contact author has declared that none of the authors has any competing interests.



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
