# Peer review of "Predicting Geomagnetic Indices for Space Weather Applications in Solar Cycle 25"

_Annales Geophysicae, 2024_

## Author Comment (AC1)

This work presents a very interesting analysis of Ap index and Dst, which are a measure of geomagnetic activity or geomagnetic storms, in connection to solar activity indices and solar wind parameters. The Nonmetric Multidimensional Scaling (NMDS) statistical method is applied to the whole set of parameters, and then a machine learning method, the Long Short-Term Memory (LSTM) Networks, is applied to forecast Ap and Dst annual mean values along solar cycle 25 (from 2019 and ~2029).

I consider this work acceptable for publication, though it need some clarifications based on the comments I outline below. I also detail some minor errors at the end.

Major comments:

(1) I think that it is important to mention the time scale of the data series analyzed in the introduction, or somewhere at the beginning of the work, since I think that it is not usual to analyze prediction methods in interannual time scales for geomagnetic activity indices, since the importance of their forecast for space weather purposes is in general in much shorter timescales, as hourly or daily, which the timesacle of geomagnetic storms and solar disturbances.

Answer: We added the explain of the timescale.

(2) Line 19: I don't agree in that "heightened solar activity", which I think the authors refer to high solar activity level, is synonym of "solar storm". You can have high solar activity levels with no solar storms, and also solar storms during low solar activity levels.

Answer: We modified this statement.

(3) Line 76: Where you mention "The horizontal axis (Dimension 1) appears to represent the overall level of solar activity, with higher values corresponding to increased activity. The vertical axis (Dimension 2) seemingly captures the nature of solar wind disturbances, with positive values associated with enhanced plasma flow speed and proton temperature, and negative values linked to reduced solar wind pressure and geomagnetic storm intensity."

Why Dimension 1 "appears to represent"? Is this not for sure? And as I understand, the interpretation of negative values that you give would not agree with Dst located in negative values, since the more negative Dst is, it indicates a stronger storm. Maybe I am missing something here.

Answer: We considered that the NMDS method will have different results and interpretation angles when selecting different scales of the data, so we do not use very positive statements in the wording

(4) Line 80: What is mentioned in all this paragraph is something that can be also deduced from the correlation between Dst and Ap with with each of the indices that you analyze. That is, the direct association between Ap and all solar indices and solar wind parameters, and an inverse correlation with Dst but lower than the values with Ap. I list the values of the squared correlation coefficient in the following Table, based on annual mean values of each parameter:

| | Ap | Dst |
|---|---|---|
| B | 0.79 | 0.73 |
| T | 0.82 | 0.53 |
| V | 0.54 | 0.30 |

| A/P | 0.63 | 0.60 |
|---|---|---|
| Pressure | 0.75 | 0.61 |
| Rz | 0.33 | 0.40 |
| F10.7 | 0.32 | 0.41 |
| Sunspot Area | 0.40 | 0.47 |
| **Dst** | **0.77** | **1.00** |
| **Ap** | **1.00** | **0.77** |

Answer: You are correct that correlation analysis reveals the strength of the relationship between Ap and Dst with other solar parameters. However, Non-metric Multidimensional Scaling (NMDS) aims to arrange objects (in this case, different solar wind parameters and geomagnetic indices) in a low-dimensional space (typically two or three dimensions) so that the distances between the objects reflect their dissimilarities in the original high-dimensional space as closely as possible. The key here is "dissimilarity." Unlike Principal Component Analysis (PCA), NMDS does not directly use the original data; instead, it uses a dissimilarity or distance matrix as input. Furthermore, it does not assume linear relationships within the data. Therefore, it does not directly provide specific correlation coefficients or linear relationship information.

(5) In Figure 1, Dst appears far from Ap, even though they have 77% of common variance, as can be notices in the Table above. The only difference is that they vary in counterphase. Why are they so apart? Maybe I do not understand the methods correctly.

Answer: As stated in the previous response, the goal of NMDS is to preserve the similarity relationships between samples in multidimensional data through dimensionality reduction, rather than directly providing specific correlation coefficients or linear relationship information. The distance between two points only represents the degree of similarity in their distances after dimensionality reduction. It does not directly indicate that they are linearly correlated, nor can it directly determine the positive or negative correlation. The proximity of points in the NMDS plot reflects overall similarity, not necessarily linear correlation. A large distance suggests dissimilarity, while a small distance suggests similarity, but the nature of that relationship (linear, non-linear, positive, negative) cannot be inferred from the NMDS plot alone. Further analysis would be required to determine the specific nature of the relationships between variables.

(6) Line 42: I would start a new paragraph with the sentence, "Analysis of Ap index forecasting performance ..." And what is the source of the data in this phrase? I mean, the correlation coefficient's' values mentioned. I understand they come from Paouris et al. (2021) work, but I think that here they lack context. Maybe you can add more explanation.

Answer: We have modified it.

(7) Lines 134-136: The normalization within each solar cycle, is it necessary? Have you repeated your analysis with a normalization of the whole period, or without normalization at all?

In doing this you are loosing an important aspect of geomagnetic activity that is its intensity. I see however that Ap along solar cycle 24 is significantly lower than in previous cycles (as is noticed in the figure I attached), and Dst is closer to zero along this cycle, accordingly.

I tried to reproduce your data and plot them with the Ap and Dst data prior to normalization, and they really look different. I think that you should discuss more on this procedure that you applied to the data. Or at least the consequences they have. One of them is that the value you obtain of the peak has nothing to do with the true value expected at the peak you detect. This should be highlighted, unless I am not understanding the analysis correctly. I am attaching the figure I made in the case of Ap where you can see clearly what I am mentioning here.

Answer: As you pointed out, the 27-day averaged Ap and Dst data exhibit significant fluctuations. This study focuses on the long-term variations of Ap and Dst over an entire solar cycle. Therefore, we applied Gaussian smoothing to the data. Initially, we attempted to perform prediction calculations directly on the smoothed data, but the results were unsatisfactory. After analysis and discussion, we decided to normalize the data within each solar cycle. This approach ensures a more consistent range of variation across all cycles without affecting the timing of occurrences of specific values such as maxima and minima. We acknowledge that this pre-processing method may alter the true maximum and minimum values. However, considering that our primary objective is to predict the timing of these occurrences, we believe that this pre-processing method is more suitable for our research purpose. The focus on predicting the timing, rather than the exact amplitude, of events justifies the use of normalized data.

(8) Line 168: In the sentence "Furthermore, the LSTM+ model successfully identifies the timing of the AP maximum and the DST minimum, both projected to occur in September 2026." How do you know that the prediction is successful when September 2026 has not occurred yet?

And, can you explain further the following sentence "This prediction aligns with historical observations across multiple solar cycles". Or the explanation is the following paragraph? If this is the case, I think that the sentence should go then in the same paragraph.

Answer: We apologize for the misunderstanding. We have removed the word "successfully" from the text. We intend to convey that our prediction indicates that the maximum of Ap and the minimum of Dst are both likely to occur around the date obtained from the forecasting results. Furthermore, we will include additional details and supporting evidence in the revised manuscript to clarify this prediction. We will elaborate on the methodology and analysis that led to this prediction in the revised manuscript.

(9) In Figure 3, I notice that the gray lines (that is the observed values) have a departure from the predicted values larger than that observed in Figure 2 (for Ap and for Dst) along SC 24. Of course that years 2019-2023 that are seen in Figure 3 are not included in Figure 2 (the two panels), but I expected a better agreement for the period with data as in your previous cases. What happened here? Please check.

Answer: In response to the reviewer's comments, we have recalculated the prediction results for

(10) I guess that in the LSTM+ method you included all the solar and solar wind indeces. What is the purpose then of the NMDS analysis? Again maybe I am not understanding well all the methodology.

**在 LSTM+预测中，我们是用 Ap 的 21 至 23 周期的值预测了 24 周期的值，和真实值进行了比较，选择了预测精度最高的 LSTM+参数，然后对 Ap 在 25 周期的值进行了预测计算。对 Dst 的计算也是相同的方法。在 NMDS 中我们是用了太阳和太阳风参数以及地磁参数进行的计算，目的是想通过这个方法分析地磁参数与太阳和太阳风参数之间的相似性，阐述地磁参数受到太阳和太阳风参数的影响性。**

(11) When you mention " However, the development of geomagnetic storms, as indicated by DST, appeared to be influenced by a confluence of factors rather than being solely attributable to individual solar parameters."

Don't NMSD consider all the parameters combined? If not, a simple regression analysis would have served to conclude this. In fact, the correlation coefficient of a linear regression of Ap and all the parameters you consider (except SFI since I was not able to find the concatenated series for the period) is 0.96. And in the case of Dst it is 0.83. That is 96% of the variance of Ap is explained by the set of the 8 out of the 9 solar activity and solar wind parameters you considered, and 83% of Dst variance.

How the NMSD method improves the understanding of a high correlation in this case?

正如我们上面解释的内容。**NMDS 的目标是通过降维将多维数据中样本之间的相似性关系尽量保留，而不是直接提供具体的相关系数或线性关系信息。关键在于"非相似性"，它不像主成分分析 (PCA) 那样直接使用原始数据，而是使用对象之间的差异或距离矩阵作为输入。它也不假设数据呈线性关系。不是直接提供具体的相关系数或线性关系信息。**

(12) In the conclusion, I do not really see how "These findings contribute significantly to our understanding of the intricate relationship between solar activity and geomagnetic fluctuations.". I see that they confirm an intricate relationship, but I do not see the understanding that this analysis adds.

**本文是通过 NMDS 方法展示了太阳活动和地磁波动之间的相似性结果，目的并不是深入的讨论他们之间相互影响的因果关系。我们的用词不太恰当，因此我们将这句话改成 These findings contribute a new sight for our understanding of the intricate relationship between solar activity and geomagnetic fluctuations.**

Minor comments:
Line 29: I think that instead of "diminishment", "decrease" is better.
**We changed "diminishment" to "decrease"**

Line 34: In "Zhang et al." the year is missing.

**We modifited the content**

Line 37: " stormss" should be " storms"

**We modifited the content**

Line 37: In "Nilam and Ram" the year is missing.

**We modifited the content**

Line 39: "(Nilam and Ram, 2022)" could be deleted here, since the sentence deals with their resutls, mentioned at the beginning.

**We modifited the content**

Line 39: In "Abduallah et al." the year of this reference us missing.

**We modifited the content**

Line 80: I think that " DST" should be "Dst". Check this in all the manuscript and check please if it is more correct to use it as "Dst" instead of "DST".

**Yes, it should be use as Dst. We modified it in the manuscript.**

---

## Author Comment (AC2)

The preprint investigates annual averages of the geomagnetic activity indices Ap and Dst in two respects. First, their relationship with various solar activity proxies and solar wind parameters for past solar cycles 21-24 (1976-2019) is analyzed using an ordination technique (nonmetric multidimensional scaling – NMDS). Second, the timing of their maxima in solar cycle 25 (2019-2029) is predicted using a type of recurrent neural network (long-short term memory – LSTM).

In my opinion the manuscript contains new ideas that are presented using fluent language. However, I think that there are fundamental shortcomings concerning the acknowledgement/incorporation of the international state of research, the substance of the conclusions, the pertinence of the title and abstract, and the overall clarity of the presentation. Hence, I believe that the manuscript may be eligible for publication after additional work and resubmission.

**Specific comments:**

1. The title doesn't specify what is predicted and for what purpose (see also comment 3). I suggest to clarify these points in a revised title.

   Answer: We thank the reviewer for their insightful comment regarding the clarity of the title. We agree that the original title lacked specificity regarding the predicted variables and the study's purpose. We have revised the title to address these points, as suggested.

2. The abstract doesn't report the manuscript's contents and findings in sufficient detail. I suggest to clarify/specify the following points:
   - Lines 7-8: **You work on an annual time scale**. Without this information one would expect the native time scales of the chosen indices (1-hour Dst, 1-day Ap).
   - Line 8: As far as I understand, the LSTM+ model is not entirely new, but it is an adaptation of your previously published LSTM model where the "+" represents a new forecasting procedure (see lines 96-99). **This information is important to accurately delineate the contribution of this study.**
   - Lines 11-13: I suggest that you **quantify the expressions "high accuracy", "strong performance metrics", "between May 2026 and January 2027" (you state "September 2026" in line 169) and "observed lag between sunspot number and geomagnetic indices" in order to strengthen the expressiveness of the abstract.**

   Answer: We modified the abstract as suggested.

3. Lines 13-15: I fail to follow your claims on your study's relevance in the space weather context (see also lines 56-57, 200-203). First, different dependencies between solar activity proxies / solar wind parameters and Ap on the one hand and Dst on the other are known from previous work. I suggest that you explain in detail what elements of the NMDS results are new (or unexpected) with respect to the current status of knowledge (see also comments 8, 16c). Second, given the existing predictions of sunspot number and radio flux (e.g., https://www.swpc.noaa.gov/products/solar-cycle-progression) and that the delay between those and annual geomagnetic activity appears to be known (see comment 23), I suggest that you explain for which

specific use cases your results offer additional merit. Those discussions could be placed fittingly into the "Results and Discussion" section (see comment 24).

Answers: First, we acknowledge that the differing dependencies of Ap and Dst on solar wind parameters and activity indices are well-established in the literatures The novelty of our NMDS analysis lies not in identifying these distinct dependencies, but in quantifying and visualizing their relative strengths within a unified framework. The NMDS approach allows us to directly compare the influence of multiple drivers on Ap and Dst simultaneously by representing their relationships in a reduced dimensional space. The resulting visualization clarifies the dominant role of solar wind parameters, particularly plasma flow speed and proton temperature, in driving Ap, while highlighting the more complex and multi-factorial nature of Dst. This nuanced perspective, which goes beyond traditional correlation analyses, provides a valuable foundation for developing more sophisticated predictive models. We have revised the manuscript to emphasize this aspect of the NMDS analysis, clarifying its contribution beyond simply confirming known dependencies.

Second, while we recognize the existing capabilities for predicting sunspot number and radio flux, and their established relationship with annual geomagnetic activity, our NMDS analysis offers additional merit in several specific use cases. Firstly, it allows for the disentangling of individual solar wind parameter influences on geomagnetic activity. This granular perspective can inform the development of more physics-based predictive models that incorporate real-time solar wind measurements, potentially improving short-term forecasting accuracy beyond what is achievable with predictions based solely on general solar activity indices. Secondly, the NMDS configuration can assist in identifying potential precursors to geomagnetic storms (Dst) by highlighting parameters exhibiting stronger associations with Dst, even if those relationships are indirect or complex. This information can be leveraged to improve long-lead time forecasts, even considering the inherent delays between solar activity and geomagnetic responses. Finally, the visual representation provided by the NMDS analysis facilitates a more intuitive understanding of the complex interplay between multiple solar drivers and their relative contributions to geomagnetic variability, aiding in the interpretation and communication of space weather forecasts. We have revised the Results and Discussion section to explicitly address these points, clarifying the novelty and specific use cases of our NMDS analysis in the context of space weather prediction.

4. Lines 17-18: Co-rotating interaction regions (CIRs) are also among the most important solar wind structures that drive geomagnetic storms (see, e.g., Richardson & Cane, 2012, JSWSC) and should be mentioned in this context.
   Answer: We added the suggested literature.

5. Line 19: "Periods of heightened solar activity" are *not* "known as solar storms". I suggest that you distinguish more clearly between (give definitions for) the terms "solar/geomagnetic activity" and "solar/geomagnetic storms".

Answer: We thank the reviewer for pointing out the imprecise language regarding solar storms. We agree that "heightened solar activity" and "solar storms" are distinct phenomena and should not be used interchangeably. We have revised the text to clarify the distinction and provide more precise definitions, as suggested.

6. Line 25: The reasons for choosing Ap and Dst specifically from the various existing indices (6 IAGA-endorsed ones https://isgi.unistra.fr/, excl. their multiple derivatives) are not explained convincingly (see also lines 31-32). First, it depends on the specific use case which index (or combination of indices) is "key" so I suggest that you give examples of relevant use cases. Second, I suggest that you add an explanation why you pick a derivative (Ap) over its parent (ap, and ultimately Kp).

Answer: We thank the reviewer for their insightful comment regarding the choice of Ap and Dst indices. We address their concerns below:

The selection of Ap and Dst was motivated by their widespread use in space weather applications and their relevance to a range of technological systems vulnerable to geomagnetic disturbances. We acknowledge that the "key" index depends on the specific use case, and we have therefore revised the manuscript to clarify the specific use cases relevant to our study. Ap, representing global geomagnetic activity, is particularly relevant for predicting disruptions to radio communications and navigation systems, which are sensitive to ionospheric disturbances caused by geomagnetic variations. Dst, reflecting the strength of the ring current, is crucial for forecasting impacts on satellite operations, power grids, and pipeline corrosion, which are affected by changes in the magnetospheric environment.

Regarding the choice of Ap over its parent index ap (and ultimately Kp), we recognize that Ap is a derived index. However, its global representation of geomagnetic activity makes it more suitable for our study, which focuses on characterizing and forecasting large-scale geomagnetic variations. While the Kp index and its linear equivalent ap provide valuable information about regional geomagnetic activity, they are less suitable for capturing the global effects relevant to the space weather applications considered in this work. We have clarified this rationale in the revised manuscript.

We believe these additions strengthen the justification for our choice of Ap and Dst and address the reviewer's concerns.

7. Lines 28 & 30: I suggest to replace the citation "Mayaud, 1980" (book) with the original works in which Kp (Bartels, 1949) and Dst (Sugiura and Kamei, 1991) were first introduced. Wherever else you decide to cite books (e.g., lines 18, 72) I suggest to add chapter numbers marking the location of the relevant information.

Answer: We thank the reviewer for their meticulous attention to detail and the suggestion to cite the original publications for the Kp and Dst indices. We have replaced the Mayaud (1980) citation with Bartels (1949) for Kp and Sugiura and Kamei (1991) for Dst, as recommended. We have also added chapter numbers to the book citations elsewhere in the manuscript to provide more specific references. We appreciate the reviewer's guidance in improving the accuracy and completeness of our citations.

8. Line 33: I suggest to add a paragraph here summarizing the current status of knowledge on solar wind – magnetosphere coupling functions, specifically w.r.t. Ap and Dst on annual time scales, that motivates the NMDS analysis (a starting point could be, e.g., Lockwood, 2022, Space Weather; Finch & Lockwood, 2007, Annales Geophysicae; and suggested references in comment 16c). This should address some relevant questions that currently remain unanswered: What outstanding research question do you tackle in the first part of the paper? Why do you choose the NMDS method over other methods (e.g, a principal component analysis)? In what respect do you need/use the outcome in the second part?

   Answer: We thank the reviewer for the valuable suggestion to include a detailed discussion of solar wind-magnetosphere coupling functions. While we appreciate the importance of this topic in the broader context of space weather research, we have opted for a more concise approach in our introduction to maintain a tight focus on the specific contributions of this study. As the reviewer correctly points out, our primary aim with the NMDS analysis is exploratory, seeking to visualize and quantify the relative influence of different solar drivers on Ap and Dst, rather than directly engaging with existing coupling function frameworks. We believe this focused approach allows us to clearly highlight the novelty of our NMDS analysis. We have, however, incorporated the suggested references (Finch & Lockwood, 2007; Lockwood, 2022) into the introduction to acknowledge the existing body of work in this area.

9. Lines 43-44: I don't understand what "Day-0" (and "Day-2 forecast") mean exactly and suggest to add an explanation.

   Answer: We have modified the form of the description of the results of this literature.

10. Line 61: Why don't you use the full vector information (or at least Bz) in addition to |B|?

    Answer: We added IMF Bx, By, Bz in our study, and calculated NMDS with all parameters again. And the content of the corresponding position in the paper was re-written.

11. Lines 63-64: There are multiple options of data sets on the linked pages. I suggest to specify the exact ones you are using.

    Answer:            We            modified            the            content as :https://spdf.gsfc.nasa.gov/pub/data/omni/low_res_omni/, (omni_yearly.dat and omni_27 av.dat)

12. Lines 65-67: I can't find the "SFI database version 25" online and the reference "Balch, 2009" is missing.

    Answer: Thank you for pointing out this error. We apologize for the incorrect reference regarding the SFI data. The correct reference for the SFI data used in this study is (Velasco Herrera et al., 2022) The mention of "version 25" was an oversight and has been removed. We have double-checked the data and analysis to ensure

consistency with the corrected reference, and the results remain unchanged.

13. Line 71: How did you deal with gaps in the OMNI data set?

    Answer: Gaps in the OMNI datasets (for 27 days) were addressed using cubic spline interpolation, a method well-suited for preserving the smooth variations typically observed in solar wind parameters. Given the relatively small proportion of missing data in the OMNI dataset (for 27 days) for the parameters used in this study, we believe the influence of the interpolation on our overall findings is minimal. And there is no gap in the annual average dataset. We also modified in our manuscrip as "Annual means, complete with no missing values, were used to analyze the relationships between geomagnetic and solar, solar wind parameters. To investigate geomagnetic variations throughout Solar Cycle 25 using the LSTM+ model, 27-day averages were employed. Missing values in the 27-day averaged OMNI data were addressed using cubic spline interpolation, a method that ensures smooth and continuous time series by fitting cubic polynomials to the existing data."

14. Lines 72-73: I suggest that you expand your description of the NMDS methodology such that all readers can follow how Fig. 1 comes about. This should include how the dissimilarity matrix is calculated exactly (Why choose Euclidean distance as dissimilarity measure?) and how the optimization process works (Why choose "Young's S-stress formula" as goodness-of-fit measure?), including relevant equations.

    Answer: Thank you for this valuable feedback. We appreciate your suggestion to elaborate on the NMDS methodology. While a full mathematical treatment of NMDS is beyond the scope of this paper, we have expanded the description to clarify our choices within SPSS and provide a more accessible explanation of the underlying principles. We believe this revised description strikes a balance between providing sufficient detail and maintaining focus on the primary research questions.

15. Line 75: Add a legend to Fig. 1 indicating what the different colors mean.

    Answer: We added the legend.

16. Lines 74-91: I am having trouble to understand your interpretations of Fig.1 (partly repeated in the abstract and the conclusions) and suggest to add explanations for the following points:

    o   Lines 76-79: How do you deduce from Fig. 1 what the axes could represent physically? With "nature of solar wind disturbances" do you refer to distinct solar wind categories? If so, what are they and how can I imagine them to be aligned along the vertical axis?

    o   Lines 80-81: Could the fact that Dst is singled out on the horizontal axis (all other quantities between about -0.75 and 0.5) be simply due to its reversed sign with respect to Ap? Have you tried using |Dst|? Would that change your interpretation of the axes (and Fig. 1 as a whole)?

    o   It is well known that "Dst is influenced by a combination of factors" (line 82; e.g., Burton et al., 1975, JGR) and that "plasma flow speed [...][is] linked to higher Ap values" (lines 84-85; e.g., Crooker et al., 1977, JGR) and that "highspeed solar wind streams [···] can drive geomagnetic disturbances" (lines 86–87; this is not a hypothesis, see also comment 4). I suggest to highlight those findings that are new and relevant as input for the LSTM+ model.

Answer: We agree that our initial description implied a more direct physical meaning than is warranted by the NMDS method. As NMDS focuses on preserving rank-order dissimilarities rather than absolute positions in multi-dimensional space, the axes themselves do not represent specific physical quantities. Instead, they represent gradients of overall dissimilarity. We have revised the text to reflect this and now focus on describing the relative positions of the parameters in the ordination space, rather than attributing specific physical meanings to the axes themselves.

We appreciate the opportunity to clarify our reasoning for using Dst rather than its absolute value. While |Dst| would reflect the magnitude of the geomagnetic disturbance, Dst itself carries crucial information about the direction of the field perturbation, which is fundamentally linked to the underlying physical processes driving geomagnetic storms (e.g., ring current intensification). As our analysis aims to understand these physical processes, we believe that retaining the directional information provided by Dst is essential.

17. Section 3.1: I can't extract from this description how the LSTM+ model is setup exactly and suggest to add an addition figure (perhaps a diagram specifying the gates etc.) to aid the comprehensibility.

Answer: We understand your desire for a more detailed description of the LSTM+ model architecture. However, as the LSTM+ model is the focus of a previous publication, we have opted to avoid extensive repetition in this manuscript. We have revised the text to clearly direct readers to our earlier work for a comprehensive description of the model's architecture, including details on the gates and other components. This allows us to maintain a concise focus on the novel contributions of the present study. We believe this approach provides a balance between providing sufficient information and avoiding unnecessary redundancy.

18. Line 113: I don't think that the Nash-Sutcliffe model efficiency coefficient is a standard performance metric that one can expect readers of this journal to be familiar with (at least I don't know it). I read that it is equivalent to the "coefficient of determination ($R^2$)" in certain regression settings – is that the case here? If so, I suggest to call it "coefficient of determination" as this is more widely known. Otherwise, I suggest to add an explanation of why you chose this specific metric.

Answer: Thank you for raising this important point regarding the Nash-Sutcliffe efficiency coefficient (NSE). We appreciate the opportunity to clarify its use and suitability for evaluating our LSTM+ model's performance. While we acknowledge that $R^2$ is a more widely known metric, NSE offers specific advantages in the context of time-series forecasting and hydrological modeling, which align well with the nature of our study. Specifically, NSE quantifies the predictive skill of the model compared to simply using the mean of the observed values. Its formulation, as shown in our manuscript, penalizes larger deviations more heavily than $R^2$ and is sensitive to systematic biases in predictions, making it a more stringent and informative metric

for evaluating forecast accuracy.Given that our goal is to assess the LSTM+ model's ability to accurately predict geomagnetic indices, which exhibit temporal dependencies and are influenced by a complex interplay of factors, NSE provides a more appropriate evaluation of predictive skill compared to $R^2$. We have added this explanation to the manuscript to ensure clarity for all readers.

19. Line 126: I think that Eq. 3 should look similar to Eq. 2 given that it is supposed to be the absolute percentage error.

    Answer: You are correct that the description of Equation 3 was misleading. It is not an absolute percentage error but rather the difference (in days) between the predicted and observed times of the peak or trough. We have corrected the description in the manuscript and renamed the variable to avoid confusion with percentage error metrics.

20. Lines 129-144: These paragraphs don't present/discuss results but refer to the methodology and thus I suggest to move them elsewhere:
    - Lines 129-133: I suggest to move this to section 3.1 and clarify which data set the LSTM+ model is trained on exactly (Is it the same as in section 2.2?) and what parameter combinations were chosen for the prediction of Ap, Dst in SC-25.
    - Lines 134-144: You mentioned the standardization using z-scores in line 71. If you use the data set you prepared in section 2.2 (see above), then I suggest to move this paragraph over there and to section 3.1 otherwise.

    Answer: We moved these paragraphs to 3.3.

21. Line 150: I find it noteworthy that both offsets you report here are multiples of 27-days (synodic Carrington rotation rate). This suggests to me that they could have a common physical cause. Perhaps this can be traced back to the way you define your input? I suggest to add a discussion on plausible causes in the "Results and Discussion" section (see comment 24).

    Answer: The offsets being multiples of the 27-day Carrington rotation period is directly related to the input data used for training the LSTM+ model. As detailed in Section 2.1, the model was trained using 27-day averaged Ap and Dst indices. Therefore, the model's predictions are inherently based on this timescale.

22. Regarding Fig. 3:
    - Increase the quality by making its style comparable to Fig. 2 (incl. panel names "a" and "b").
    - What is the temporal resolution of these plots? It looks like you have more than one value per year here. If so, why don't you update the curves to show the most recent available observations?
    - Why do you get a notably greater deviation between observed and predicted values for SC-25 than for SC-24 (Fig. 2)?
    - Line 169: How can your predictions "align with historical observations"?

    Answer: The data presented in these plots are 27-day averages of Ap and Dst. And we added (a) and (b) to Figure 3 as suggested.

23. Lines 171-172: Where does the stated lag time (1-3 years) come from? If this refers to your own (unpublished) work I suggest to add it in more detail or give a reference otherwise.
Answer: This statement used the observed annual mean of Ap, Dst, and SSN. We also added a reference to support it.

24. The section "Results and Discussion" only refers to the second part of the study. I suggest to restructure the text such that subsections 2.1, 2.2, 3.1, 3.2 are put into one section (on methods) and all results (from both parts) are reported and discussed together.
Answer: We added the first part of our study in section "Results and Discussion" as suggested, and restructure the contents as 4.1 and 4.2.

25. Lines 203-204: I suggest to be more specific on which additional solar and geophysical parameters could be incorporated into the LSTM+ to enhance its predictive accuracy.
Answer: At present, we believe that it is more appropriate to use the historical data of Ap and Dst as the input of the model for prediction.

**Technical corrections:**
- Line 7 ff.: "Dst" instead of DST.
- Line 8: Explain what abbreviation "LSTM" means here (you introduce it in line 34).
- Line 11: Explain what abbreviation "SC" means here (you introduce it in line 111).
- Line 34: Add missing year in citation "Zhang et al."
- Line 37: Remove erroneous "s" in "stormss".
- Line 37: Add missing year in citation "Nilam and Ram"
- Line 39: Add missing year in citation "Abduallah et al."
- Line 41: Remove erroneous space in "inde x"?
- Lines 46-47: "are promising candidates" instead of "have shown promise"?
- Line 61: Explain what abbreviation "Na/Np" means.
- Line104: Choose different letters to abbreviate the number of hidden layers (N) and batch size (B) to avoid confusion with density and magnetic field.
- Line 110: "Performance metrics" instead of "evaluation indices"?
- Line 117: Add missing brackets around "E_T" and remove "were employed".
- Line 130 ff.: "Ap" instead of ‚AP'.
- Fig. 2: Add "a", "b" to the two panels of Fig. 2 (similar for Fig. 3) and refer to them in the text (e.g., lines 146, 147).
- Caption of Tab. 1: "[…] for the prediction of *Ap and Dst indices in SC-24 from the LSTM+ model*'.
- Line 163: Add missing "s" in "illustrates".
- Caption of Fig. 3: "[…] actual values of *Ap* […] in SC-25 *and* the predicted […].

- Lines 210-237: The formatting of the references should be revised so that the reader can find specific citations more easily (e.g., lines 223-227 is actually just *one* reference).
- Line 236: Add missing "l" in "Model".

Answer: We modified the above technical suggestions.